# Pooling is neither necessary nor sufficient for appropriate deformation stability in CNNs

## Abstract

Many of our core assumptions about how neural networks operate remain empirically untested. One common assumption is that convolutional neural networks need to be stable to small translations and deformations to solve image recognition tasks. For many years, this stability was baked into CNN architectures by incorporating interleaved pooling layers. Recently, however, interleaved pooling has largely been abandoned. This raises a number of questions: Are our intuitions about deformation stability right at all? Is it important? Is pooling necessary for deformation invariance? If not, how is deformation invariance achieved in its absence? In this work, we rigorously test these questions, and find that deformation stability in convolutional networks is more nuanced than it first appears: (1) Deformation invariance is not a binary property, but rather that different tasks require different degrees of deformation stability at different layers. (2) Deformation stability is not a fixed property of a network and is heavily adjusted over the course of training, largely through the smoothness of the convolutional filters. (3) Interleaved pooling layers are neither necessary nor sufficient for achieving the optimal form of deformation stability for natural image classification. (4) Pooling confers *too much* deformation stability for image classification at initialization, and during training, networks have to learn to *counteract* this inductive bias. Together, these findings provide new insights into the role of interleaved pooling and deformation invariance in CNNs, and demonstrate the importance of rigorous empirical testing of even our most basic assumptions about the working of neural networks.

## 1 Introduction

Within deep learning, a variety of intuitions have been assumed to be common knowledge without empirical verification, leading to recent active debate (Rahimi & Recht, 2017; LeCun, 2017; Sculley et al., 2015; 2018). Nevertheless, many of these core ideas have informed the structure of broad classes of models, with little attempt to rigorously test these assumptions.

In this paper, we seek to address this issue by undertaking a careful, empirical study of one of the foundational intuitions informing convolutional neural networks (CNNs) for visual object recognition: the need to make these models stable to small translations and deformations in the input images. This intuition runs as follows: much of the variability in the visual domain comes from slight changes in view, object position, rotation, size, and non-rigid deformations of (e.g.) organic objects; representations which are invariant to such transformations would (presumably) lead to better performance. This idea is arguably one of the core principles initially responsible for the architectural choices of convolutional filters and interleaved pooling LeCun et al. (1998; 2015), as well as the deployment of parametric data augmentation strategies during training Simard et al. (2003). Yet, despite the widespread impact of this idea, the relationship between visual object recognition and deformation stability has not been thoroughly tested, and we do not *actually know* how modern CNNs realize deformation stability, if they even do at all.

Moreover, for many years, the very success of CNNs on visual object recognition tasks was thought to depend on the interleaved pooling layers that purportedly rendered these models insensitive to small translations and deformations. However, despite this reasoning, recent models have largely abandoned interleaved pooling layers, achieving similar or greater success without them Springenberg et al. (2014); He et al. (2016).

These observations raise several critical questions. Is deformation stability necessary for visual object recognition? If so, how is it achieved in the absence of pooling layers? What role does interleaved pooling play when it is present?

Here, we seek to answer these questions by building a broad class of image deformations, and comparing CNNs' responses to original and deformed images. While this class of deformations is an artificial one, it is rich and parametrically controllable, includes many commonly used image transformations (including affine transforms: translations, shears, and rotations, and thin-plate spline transforms, among others) and it provides a useful model for probing how CNNs might respond to natural image deformations. We use these to study CNNs with and without pooling layers, and how their representations change with depth and over the course of training. Our contributions are as follows:

- Networks without pooling are sensitive to deformation at initialization, but ultimately learn representations that are stable to deformation.
- The inductive bias provided by pooling is too strong at initialization, and deformation stability in these networks decrease over the course of training.
- The pattern of deformation stability across layers for trained networks with and without pooling converges to a similar structure.
- Networks both with and without pooling implement and modulate deformation stability largely through the smoothness of learned filters.

More broadly, this work demonstrates that our intuitions as to why neural networks work can often be inaccurate, no matter how reasonable they may seem, and require thorough empirical and theoretical validation.

## 1.1 Prior work

**Invariances in non-neural models.** There is a long history of non-neural computer vision models architecting invariance to deformation. For example, SIFT features are local features descriptors constructed such that they are invariant to translation, scaling and rotation Lowe (1999). In addition, by using blurring, SIFT features become somewhat robust to deformations. Another example is the deformable parts models which contain a single stage spring-like model of connections between pairs of object parts giving robustness to translation at a particular scale Felzenszwalb et al. (2008).

**Deformation invariance and pooling.** Important early work in neuroscience found that in the visual cortex of cats, there exist special complex-cells which are somewhat insensitive to the precise location of edges Hubel & Wiesel (1968). These findings inspired work on the neocognitron, which cascaded locally-deformation-invariant modules into a hierarchy Fukushima & Miyake (1982). This, in turn, inspired the use of pooling layers in CNNs LeCun et al. (1990). Here, pooling was directly motivated as conferring invariance to translations and deformations. For example, LeCun et al. (1990) expressed this as follows: *Each feature extraction in our network is followed by an additional layer which performs a local averaging and a sub-sampling, reducing the resolution of the feature map. This layer introduces a certain level of invariance to distortions and translations.* In fact, until recently, pooling was still seen as an essential ingredient in CNNs, allowing for invariance to small shifts and distortions Simonyan & Zisserman (2014); He et al. (2016); Krizhevsky et al. (2012); Simonyan & Zisserman (2014); LeCun et al. (2015); Giusti et al. (2013).

**Previous theoretical analyses of invariances in CNNs.** A significant body of theoretical work shows formally that scattering networks, which share some architectural components with CNNs, are stable to deformations Mallat (2012); Sifre & Mallat (2013); Bruna & Mallat (2013); Mallat (2016). However this work does not apply to widely used CNN architectures for two reasons. First, there are significant architectural differences, including in connectivity, pooling, and non-linearities. Second, and perhaps more importantly, this line of work assumes that the filters are fixed wavelets that do not change during training.

The more recent theoretical study of Bietti & Mairal (2017) uses reproducing kernel Hilbert spaces to study the inductive biases (including deformation stability) of architectures more similar to the CNNs used in practice. However, this work assumes the use of interleaved pooling layers between the convolutional layers, and cannot explain the success of more recent architectures which lack them.

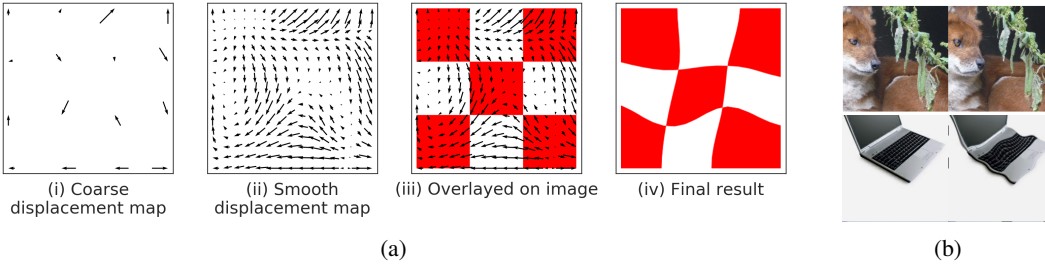

Figure 1: (a) **Generating deformed images**: To randomly deform an image we: (i) Start with a fixed evenly spaced grid of control points (here 4x4 control points) and then choose a random source for each control point within a neighborhood of the point; (ii) we then smooth the resulting vector field using thin plate interpolation; (iii) vector field overlayed on original image: the value in the final result at the tip of an arrow is computed using bilinear interpolation of values in a neighbourhood around the tail of the arrow in the original image; (iv) the final result. (b) **Examples of deformed ImageNet images.** left: original images, right: deformed images. While the images have changed significantly, for example under the $L^2$ metric, they would likely be given the same label by a human.

**Empirical investigations.** Previous empirical investigations of these phenomena in CNNs include the work of Lenc & Vedaldi (2015), which focused on a more limited set of invariances such as global affine transformations. More recently, there has been interest in the robustness of networks to *adversarial* geometric transformations in the work of Fawzi & Frossard (2015) and Kanbak et al. (2017). In particular, these studies looked at worst-case sensitivity of the output to such transformations, and found that CNNs can indeed be quite sensitive to particular geometric transformations (a phenomenon that can be mitigated by augmenting the training sets). However, this line of work does not address how deformation sensitivity is generally achieved in the first place, and how it changes over the course of training. In addition, these investigations have been restricted to a limited class of deformations, which we seek to remedy here.

## 2 METHODS

### 2.1 DEFORMATION SENSITIVITY

In order to study how CNN representations are affected by image deformations we first need a controllable source of deformation. Here, we choose a flexible class of local deformations of image coordinates, i.e., maps $\tau : \mathbb{R}^2 \to \mathbb{R}^2$ such that $\|\nabla \tau\|_\infty < C$ for some $C$, similar to Mallat (2012). We choose this class for several reasons. First, it subsumes or approximates many of the canonical forms of image deformation we would want to be robust to, including:

- **Pose:** Small shifts in pose or location of subparts
- **Affine transformations:** translation, scaling, rotation or shear
- **Thin-plate spline transforms**
- **Optical flow:** Roth & Black (2007); Rosenbaum et al. (2013)

We show examples of several of these in Section 2 of the supplementary material.

This class also allows us to modulate the strength of image deformations, which we deploy to investigate how task demands are met by CNNs. Furthermore, this class of deformations approximates most of the commonly used methods of data augmentation for object recognition Simard et al. (2003); Wong et al. (2016); Cireşan et al. (2010).

While it is in principle possible to explore finer-grained distributions of deformation (e.g., choosing adversarial deformations to maximally shift the representation), we think our approach offers good coverage over the space, and a reasonable first order approximations to the class of natural deformations. We leave the study of richer transformations—such as those requiring a renderer to produce or those chosen adversarially Fawzi & Frossard (2015); Kanbak et al. (2017)—as future work.

Throughout this work we will measure the stability of CNNs to deformations by: i) sampling random deformations as described below; ii) applying these deformations to input images; iii) measuring the effect of this image deformation on the representations throughout the various layers of the CNN.

**Generating deformed images.** We use the following method to sample image deformations from this class. For ImageNet we use a grid of 5x5 evenly spaced control points on the image and then choose a destination for each control point uniformly at random with a maximum distance of $C = 10$ pixels in each direction. For CIFAR-10 images, we used 3x3 evenly spaced control points and a maximum distance of $C = 2$ pixels in each direction. The resulting deformation map was then smoothed using thin plate spline interpolation and finally the deformation was applied with bilinear interpolation Duchon (1977). The entire process for generating deformed images is illustrated in Figure 1a.

**Measuring sensitivity to deformation.** For a representation $r$ mapping from an input image (e.g., 224x224x3) to some layer of a CNN (e.g., a tensor of size 7x7x512), we measure sensitivity of the representation $r$ to a deformation $\tau$ using the *Normalized Cosine Distance*:

$$\frac{\mathrm{d}_{\cos}(r(x), r(\tau(x)))}{\mathrm{median}(\mathrm{d}_{\cos}(r(x), r(y)))}$$

where $\mathrm{d}_{\cos}$ is the cosine distance. That is, we normalize distances by the median distance in representation space between randomly selected images from the original dataset. For our results, we average this quantity over 128 images and compute the median using all pairwise distances between the representations of the 128 images. We also found that using Euclidean distance instead of cosine distance yielded qualitatively similar results.

## 2.2 Networks

All networks trained for our experiments are based on a modified version of the VGG network Simonyan & Zisserman (2014). A detailed description of the networks can be found in Section 1 of the supplementary material.

We compared networks with the following downsampling layers in our CIFAR-10 experiments: **Subsample:** Keep top left corner of each 2x2 block. **Max-pool:** Standard max-pooling layer. **Average-pool:** Standard average-pooling layer. **Strided:** we replace the max pooling layer with a convolutional layer with kernels of size 2x2 and stride 2x2. **Strided-ReLU:** we replace the max pooling layer with a convolutional layer with kernels of size 2x2 and stride 2x2. The convolutional layer is followed by batch-norm and ReLU nonlinearity. For our ImageNet experiments, we compared only Max-pool and Strided-ReLU due to computational considerations.

To rule out variability due to random factors in the experiment (random initialization and data order), we repeated all experiments with 5 different random seeds for each setting. The error bands in the plots correspond to 2 standard deviations estimated across these 5 experiments.

## 3 Learned deformation stability is similar with and without pooling

It is a commonly held belief that pooling leads to invariance to small translations and deformations. In this section we investigate two questions: (1) Is pooling *sufficient* for achieving the correct amount of stability to deformation? (2) Is pooling *necessary* for achieving stability?

**Pooling influences deformation stability.** To test whether pooling on its own is sufficient for achieving any significant change in deformation stability, we measured the sensitivity to deformation of networks with pooling at initialization. As can be seen in Figure 2a, we find that indeed pooling leads to representations that are more stable to deformation at initialization than representations in networks without pooling. This result also provides us with a basic sanity check that our experimental setup is reasonable.

**Pooling does not determine the final pattern of stability across layers.** To what extent does pooling determine the final pattern of deformation stability? Also, if pooling leads to a suboptimal

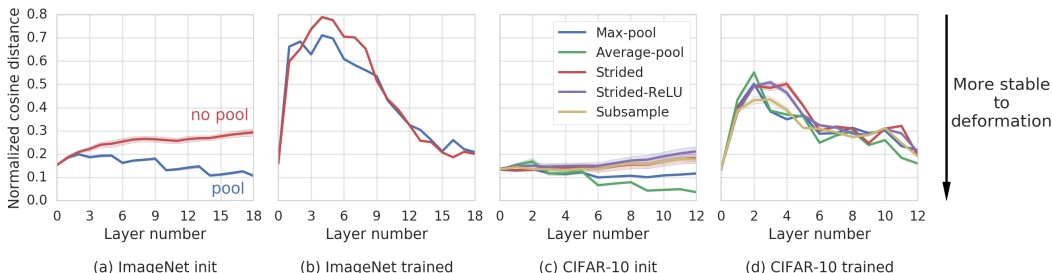

Figure 2: **Pooling confers stability to deformation at initialization but the stability changes significantly over the course of training and converges to a similar stability regardless of whether pooling is used.** (a) At initialization, networks with max-pooling are less sensitive to deformation. (b) After training, networks with and without max-pooling have very similar patterns of sensitivity to deformation throughout the layers. Similar patterns emerge for CIFAR-10: (c) At initialization, pooling has significant impact on sensitivity to deformation but (d) after training, the choice of downsampling layers has little effect on deformation stability throughout the layers. Layer number 0 corresponds to the input image; The layers include the downsampling layers; The final layer corresponds to the final downsampling layer. For CIFAR-10 we therefore have 1 input layer, 8 convolutional layers and 4 pooling layers for a total of 13 layers.

pattern of deformation stability for the task, then to what extent can learning correct for this? To test this, we measured the pattern of sensitivity to deformation of networks before and after training. Surprisingly, we found that the sensitivity to deformation of networks with pooling actually increases significantly over the course of training (Figure 2b). This result suggests that *the inductive bias for deformation stability conferred by pooling is actually too strong*, and that while deformation stability might be helpful in some cases, it is not always helpful.

**Networks with and without pooling converge to similar patterns of deformation stability across layers** If learning substantially changes the layerwise pattern of deformation stability in the presence of pooling, to what extent does pooling actually influence the final pattern of deformation stability? To test this, we measured the layerwise pattern of sensitivity to deformation for networks trained on ImageNet with and without interleaved pooling. Surprisingly, the layerwise pattern of sensitivity to deformation for networks with and without pooling was highly similar, suggesting that the presence of pooling has little influence on the learned pattern of deformation stability (Figure 2a-b).

To rule out the dependence of this result on the particular dataset and the choice of downsampling layer, we repeated the experiments on CIFAR-10 and with a variety of different downsampling layers, finding qualitatively similar results (Figure 2c-d). While the downsampling layer exerted a significant effect on deformation sensitivity at initialization, these differences had largely vanished by the conclusion of training, and all networks converged to a similar pattern.

These results help to explain the recent observation that CNNs without pooling can achieve the same or higher accuracy than networks with pooling on image classification tasks Springenberg et al. (2014). While pooling does grant some deformation stability at initialization, this inductive bias is too strong and must be removed over training, and nearly identical patterns of deformation stability can be easily learned by networks without any pooling at all.

## 4 FILTER SMOOTHNESS CONTRIBUTES TO DEFORMATION STABILITY

If pooling is not the major determinant of deformation stability, then what is? One possibility is that filter smoothness might lead to deformation stability. Informally, a smooth filter can be decomposed into a coarser filter (of similar norm) followed by a smoothing operation similar to average pooling or smoothing with a Gaussian kernel Bietti & Mairal (2017). A smooth filter might therefore function similarly to the combination of a pooling layer followed by a convolutional layer. If this is in fact the case, then CNNs with smooth filters may exhibit similar behavior to those with interleaved pooling Bietti & Mairal (2017).

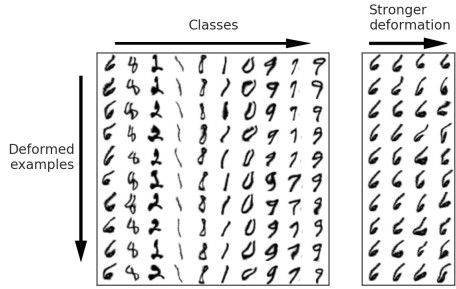

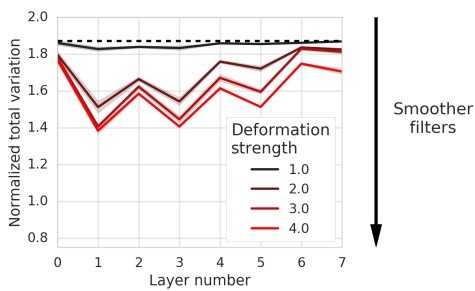

(a) Deformed templates classes for synthetic tasks

(b) Smoothness of learned filters

Figure 3: **Tasks requiring more deformation invariance lead to smoother filters.** (a) We generate synthetic tasks where each class is based on a single MNIST image and within each class examples are generated by applying a random deformation of strength $C$ to this class image. The image on the left is generated using deformations of strength 3. The columns for the image on the right are generated using deformations of strengths 1, 2, 3, 4 respectively. (b) After training, filters from networks trained on tasks where stronger deformations are used are smoother. Dotted black line indicates average value at initialization.

In this section, we demonstrate empirically that filter smoothness is critical for determining deformation stability. To do this, we first define a measure of filter smoothness. We then show that forcing filter smoothness at initialization leads to deformation stability. Next, we show that in a series of synthetic tasks requiring increasing stability to deformations, CNNs learn progressively smoother filters. Finally, we demonstrate on ImageNet and CIFAR-10 that filter smoothness increases as a result of training, even for networks with pooling.

**Measuring filter smoothness.** For a 4D (height $\times$ width $\times$ input filters $\times$ output filters) tensor $W$ representing convolutional weights we define the *normalized total variation*: $\frac{\text{TV}(W)}{\|W\|_1}$ where $\text{TV}(W) = \sum_{i,j} \|W_{i,j,\cdot,\cdot} - W_{i+1,j,\cdot,\cdot}\|_1 + \|W_{i,j,\cdot,\cdot} - W_{i,j+1,\cdot,\cdot}\|_1$ where $i$ and $j$ are the indices for the spatial dimensions of the filter. This provides a measure of filter smoothness.

For filters that are constant over the spatial dimension—by definition the smoothest possible filters— the normalized total variation would be zero. At initialization for a given layer, the expected smoothness is identical across network architectures[1]. Given that this value has an approximately fixed mean and small standard deviation across layers and architectures, we plot this as a single dotted black line in Figure 4 and Figure 3 for simplicity.

**Initialization with smooth filters leads to deformation stability.** To test whether smooth filters lead to deformation stability, we initialized networks with different amounts of filter smoothness and asked whether this yielded greater deformation stability. To initialize filters with different amounts of smoothness, we used our usual random initialization[2], but then convolved them with Gaussian filters of varying smoothness. Indeed we found that networks initialized with smoother random filters are more stable to deformation (Figure 5a), suggesting that filter smoothness is sufficient for deformation stability.

**Requiring increased stability to deformation in synthetic tasks leads to smoother filters.** The above result demonstrates that randomly-initialized smooth filters lead to greater deformation stability. The distribution of learned filters may nevertheless differ significantly from that of random filters. We therefore asked whether smoother filters are actually learned in tasks requiring stability to stronger deformation. To test this, we constructed a set of synthetic classification tasks in which each class consists of deformed versions of a single image. Each task varied the strength of deformation, $C$, used

---

[1]We estimated this average smoothness empirically by resampling filters 10,000 times and found an average smoothness of approximately 1.87 (this differed between layers only in the fourth decimal place) with a standard deviation that depended on the size of the layer, ranging from 0.035 in the first layer to 0.012 for the larger layers.

[2]Truncated normal with standard deviation $1/\sqrt{n_{in}}$, where $n_{in}$ is the number of inputs.

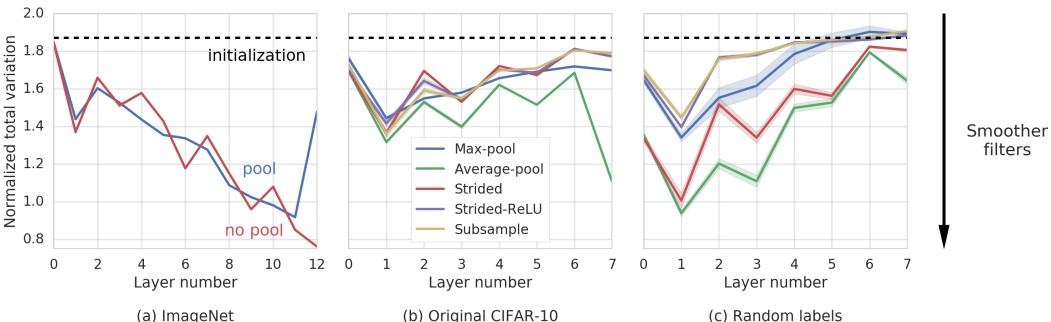

Figure 4: **Training leads to smoother filters.** (a) and (b) After training, the filters are significantly smoother and different architectures converge to similar levels of filter smoothness. (c) When training on random labels the smoothness of filters depends largely on the chosen downsampling layer. Interestingly, the smoothness of filters when training on ImageNet (a) increases from layer to layer, whereas for CIFAR-10 (b) the smoothness decreases from layer to layer. Dotted black lines indicate average value at initialization.

to generate the examples within each class (Figure 3a). We then measured the effect of increasing the intra-class deformation (i.e., increasing $C$) on the smoothness of the filters learned in each task. Consistent with our previous result, we observed that stronger deformations led to smoother filters after training (Figure 3). This result demonstrates that in synthetic tasks requiring deformation stability, the amount of learned filter smoothness directly correlates with the amount of deformation.

**Filter smoothness increases as a result of training on real datasets.** Finally, we asked whether filter smoothness increases over the course of training in more realistic datasets. To test this we examined the filter smoothness across layers for a variety of network architectures trained on both ImageNet (Figure 4a) and CIFAR-10 (Figure 4b). For both datasets and all architectures, filters become smoother over training.

Taken together, these results demonstrate that filter smoothness is sufficient to confer deformation stability, that the amount of filter smoothness tracks the amount of deformation stability required, and that on standard image classification tasks, filter smoothness is learned over the course of training.

## 5 FILTER SMOOTHNESS DEPENDS ON THE SUPERVISED TASK

In the previous section we demonstrated that smooth filters are sufficient to confer deformation stability of CNN representations, but it remains unclear which aspects of training encourage filter smoothness and deformation stability. One possibility is that smooth filters emerge as a consequence of the distribution $P(X)$ of the input images $X$. Alternatively, the nature of the supervised task itself may be critical (i.e. the conditional distribution $P(Y|X)$ of the labels $Y$ given the inputs $X$).

To test what role $P(X)$ and $P(Y|X)$ play in the smoothness of the learned filters, we followed the method of Zhang et al. (2016), and trained networks on a modified versions of the CIFAR-10 dataset in which we replace the labels with uniform random labels (which are consistent over training epochs). The representations learned under such tasks have been studied before, but not in the context of deformation stability or filter smoothness Morcos et al. (2018); Achille & Soatto (2017). Therefore, we analyzed the patterns of deformation stability and filter smoothness of networks trained on random images (modifying $P(X)$) and random labels (modifying $P(Y|X)$ but holding $P(Y|X)$ fixed).

In contrast to networks trained on the original datasets, we found that networks with different architectures trained on random labels converged to highly different patterns of deformation stability across layers (Figure 5b). These patterns were nevertheless consistent across random seeds.

This result suggests that both the architecture and the task bias the learned pattern of deformation stability, but with different strengths. In the presence of a structured task (as in Section 3), the inductive bias of the architecture is overridden over the course of learning; all networks thus converge

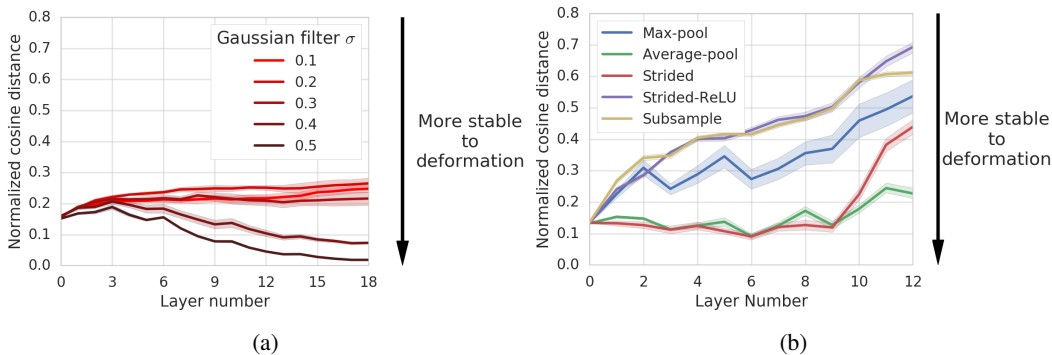

(a)                                                            (b)

Figure 5: (a) **Initialization with smoother random filters lead to deformation stability.** We smooth filters by convolving with a Gaussian filter with standard-deviation $\sigma$ and then measure the sensitivity to deformation. As we increase the smoothness of the filters by increasing $\sigma$, the representations became less sensitive to deformation. Darker lines are for smoother random filters. (b) **Deformation stability is architecture dependent when training with random labels.**

to similar layerwise patterns of deformation stability. However, in the absence of a structured task (as is the case in the random labels experiments), the inductive biases of the architecture strongly influences the final pattern of deformation stability.

## 6 DISCUSSION

In this work, we have rigorously tested a variety of properties associated with deformation stability. We demonstrated that while pooling confers deformation stability at initialization, it does not determine the pattern of deformation stability across layers. This final pattern is consistent across network architectures, both with and without pooling. Moreover, the inductive bias conferred by pooling is in fact too strong for ImageNet and CIFAR-10 classification; this therefore has to be counteracted during training. We also found that filter smoothness contributes significantly to achieving deformation stability in CNNs. Finally, these patterns remain a function of the task being learned: the joint distribution of inputs and outputs is important in determining the level of learned deformation stability.

Together, these results provide new insights into the necessity and origins of deformation stability. They also provide an instructive example of how simple properties of learned weights can be investigated to shed light on the inner workings of deep neural networks.

One limitation of this work is that we only focused on deformations sampled from a particular distribution. We also only measured average sensitivity over these deformations. In future work, it would be informative to explore similar questions but with the worst case deformations found via maximization of the deformation sensitivity Fawzi & Frossard (2015); Kanbak et al. (2017).

Finally, our work compares only two points in time: the beginning and the end of training. There remain open questions about how these characteristics change over the course of training. For example, when do filters become smooth? Is this a statistical regularity that a network learns early in training, or does filter smoothness continue to change even as network performance begins to asymptote? Does this differ across layers and architectures? Is the trajectory toward smooth filters and deformation stability monotone, or are there periods of training where filters become smoother and then periods when the filter smoothness decreases? Future work will be required to answer all of these questions.

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

## 7 MODEL ARCHITECTURES AND TRAINING

All networks we trained for our experiments are based on a modified version of the VGG network Simonyan & Zisserman (2014). The networks consist of multiple blocks as follows:

- **Conv block:** A block consists of multiple layers of convolutional filters followed by batch-norm and then a ReLU non-linearity, we will denote the structure of a block by the number of filters in each conv layer and the number of layers, for example, 2x64 will mean a block with 2 layers with 64 filters in the convolutional layers. All filters have a spatial dimension of 3x3.

- **Downsampling:** Each block is followed by a downsampling layer where the spatial resolution is decreased by a factor of 2 in both height and width dimensions.

- **Global average pooling:** we replace the fully connected layers of VGG with global average pooling and a single linear layer as is now commonly done (He et al. (2016) following Springenberg et al. (2014)).

For the ImageNet experiments, we used networks with block structure 2x64, 2x128, 3x256, 3x512, 3x512. For the CIFAR10 experiments, we used networks with block structure 2x32, 2x64, 2x128, 2x256.

We compared networks with the following downsampling layers in our CIFAR10 experiments: **Subsample:** Keep top left corner of each 2x2 block. **Max-pool:** Standard max-pooling layer. **Average-pool:** Standard average-pooling layer. **Strided:** we replace the max pooling layer with a convolutional layer with kernels of size 2x2 and stride 2x2. **Strided-ReLU:** we replace the max pooling layer with a convolutional layer with kernels of size 2x2 and stride 2x2. The convolutional layer is followed by batch-norm and ReLU nonlinearity. For our ImageNet experiments, we compared only Max-pool and Strided-ReLU due to computational considerations.

To rule out variability due to random factors in the experiment (initial random weights, order in which data is presented), we repeated all experiments 5 times for each setting. The error bands in the plots correspond to 2 standard deviations estimated across these 5 experiments.

# 8 CLASS OF DEFORMATIONS

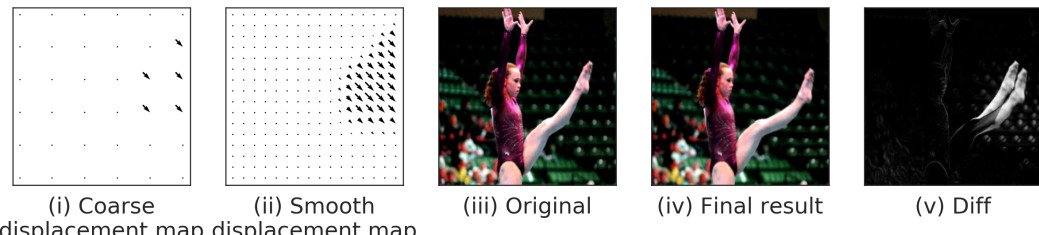

(i) Coarse displacement map   (ii) Smooth displacement map   (iii) Original   (iv) Final result   (v) Diff

Figure 6: **Changes in pose can be well approximated using the deformations we consider.**

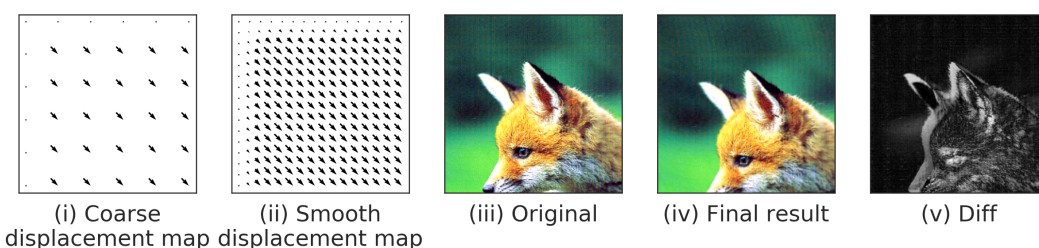

(i) Coarse displacement map   (ii) Smooth displacement map   (iii) Original   (iv) Final result   (v) Diff

Figure 7: **Translation can be well approximated using the deformations we consider.**

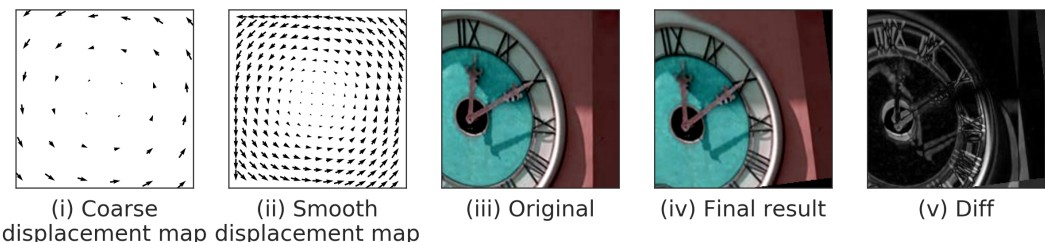

(i) Coarse displacement map   (ii) Smooth displacement map   (iii) Original   (iv) Final result   (v) Diff

Figure 8: **Rotation can be well approximated using the deformations we consider.**

In this section we give a few example deformations that approximate other geometric transformations that are often of interest such as pose, translation and rotation. Examples of approximating pose, translation and rotation are visulaized in Figures 6, 7, and 8 respectively. Note that while translation and rotation are often studied as global image transformations, the class of deformations we use can approximate applying these transformations locally (e.g., the pose example shows a local translation that could not be captured by a global affine transform).

