# OpenReview forum: "Pooling Is Neither Necessary nor Sufficient for Appropriate Deformation Stability in CNNs"
_ICLR.cc/2019/Conference_

### Official Review · AnonReviewer1 · 2018-11-08
**Ask good questions but need more insightful analysis**

**Rating:** 5
**Confidence:** 2

**Review:**

The work does an analysis of impact of different pooling strategies on image classification with deformations. It shows different pooling strategies reach to similar levels of deformation stability after sufficient training. It also offers an alternative technique with smoothness filters with to CNNs more stable.
Pros:
The paper considers a wide variety of pooling strategies and deformation techniques for evaluation.  Fair experiments with conclusion of similar stability of different pool layers after training is very evident.
Cons:
i) Results on CIFAR 10 show pooling has little effect but is it unnecessary for harder problems as well? What about human pose datasets where deformation is inherent?
iii) Although, the results presented on smoother filter initialization are interesting, but these are results are not compared in a one to one setting to different pooling methods, convolutions or residual networks.

This paper tries to argue that pooling is unnecessary for deformation invariance, as title suggests, and proposes initialization based on smooth filters as an alternative. Results are presented on CIFAR 10 to show the same, albeit on a trained network. However, CIFAR 10 is not a difficult dataset and the level of cosine sensitivity (shown as same with and without pooling) could very well be a steady state for the specific classification task. Imagenet dataset doesn't seem to show ablative studies. So this little evidence is insufficient to conclude that pooling is unnecessary.  Also as mentioned in the conclusion of the paper, the effect of pooling through the course of training would add more weight.

---

### Official Review · AnonReviewer1 · 2018-11-08
**Ask good questions but need more insightful analysis**

**Rating:** 5
**Confidence:** 2

**Review:**

The work does an analysis of impact of different pooling strategies on image classification with deformations. It shows different pooling strategies reach to similar levels of deformation stability after sufficient training. It also offers an alternative technique with smoothness filters with to CNNs more stable.
Pros:
The paper considers a wide variety of pooling strategies and deformation techniques for evaluation.  Fair experiments with conclusion of similar stability of different pool layers after training is very evident.
Cons:
i) Results on CIFAR 10 show pooling has little effect but is it unnecessary for harder problems as well? What about human pose datasets where deformation is inherent?
iii) Although, the results presented on smoother filter initialization are interesting, but these results are not compared in a one to one setting to different pooling methods, convolutions or residual networks.

This paper tries to argue that pooling is unnecessary for deformation invariance, as title suggests, and proposes initialization based on smooth filters as an alternative. Results are presented on CIFAR 10 to show the same, albeit on a trained network. However, CIFAR 10 is not a difficult dataset and the level of cosine sensitivity (shown as same with and without pooling) could very well be a steady state for the specific classification task. Imagenet dataset doesn't seem to show ablative studies. So this little evidence is insufficient to conclude that pooling is unnecessary.  Also as mentioned in the conclusion of the paper, the effect of pooling through the course of training would add more weight.

---

> ### Author Response · Authors · 2018-11-26
> **Thank you for your feedback**
>
> Thank you for your kind feedback.
>
> “This paper tries to argue that pooling is unnecessary for deformation invariance.”
> Perhaps we should have made this clearer in our writing, but our claim is not that pooling is *never* necessary. Instead, our claim is that pooling is not *always* necessary and that there is an alternative mechanism that can lead to stability to deformation, namely smooth filters. Further, we show that on very commonly studied tasks, this mechanism is at play. Perhaps our choice of title lead to some confusion, but we were trying to say pooling is “not necessary” by which we meant “not required” rather than “uneccessary” which seems to imply “never helpful”. We believe these are the common uses of these terms but perhaps we should have chosen a different title or made our assertion clearer.
>
> “Results on CIFAR 10 show pooling has little effect but is it unnecessary for harder problems as well? What about human pose datasets where deformation is inherent?”
> Thank you for this suggestion, this indeed is an interesting question. However, at the same time, we do not believe this question needs to be answered to establish our main point in this paper. Our point is NOT that pooling is never helpful. Our point is that for tasks that benefit from deformation stability, it is possible to learn to be stable to deformation by learning smooth filters. We also show that this mechanism is at play on two of the most commonly studied computer vision tasks in machine learning.
>
> “the level of cosine sensitivity (shown as same with and without pooling) could very well be a steady state for the specific classification task.”
> It would be very helpful to us if you could expand on this statement. Are you claiming that perhaps deformation stability is merely correlated with good performance at the end of training rather than causing it?
>
> “Also as mentioned in the conclusion of the paper, the effect of pooling through the course of training would add more weight.”
> Thank you for the encouragement to pursue this line of work!

---

> > ### Comment · AnonReviewer1 · 2018-11-27
> > **Replying to additional clarifications**
> >
> > "Perhaps we should have made this clearer in our writing, but our claim is not that pooling is *never* necessary. Instead, our claim is that pooling is not *always* necessary and that there is an alternative mechanism that can lead to stability to deformation, namely smooth filters."
> >
> > Thanks for clarifying your assertion! However, 'Pooling is neither necessary nor sufficient for deformation stability' is a very strong claim. 'Pooling is not always necessary and smooth filters can sometimes be used for deformation stability' shows little contribution and would require an air-tight analysis to be considered above acceptance. When does pooling fail? When does smooth filter initialization outperform pooling? When does smooth filter initialization fail? I do not see a one-one comparison between smooth filters with different pooling methods. Figure 5a comes close but it doesn't compare to average pooling and it seems to be on a different task as compared to Figure 2d. Figure 5b is misleading when looked alongside Figure 5a as they deal with totally different tasks. Also, the purpose of pooling is more than just deformation invariance and a comparison of how other properties of network are affected by replacing pooling with smooth filters is also necessary to reach a conclusion, such as the downstream performance of discriminator.
> >
> > "It would be very helpful to us if you could expand on this statement."
> >
> > You have understood the comment correctly. Given the network (VGG) and training data (CIFAR 10), in figure 2d, the cosine sensitivity may be a property of the training task i.e. a low training error could be correlated to cosine similarity of around 0.2. For a similar training error, how does the smooth filter initialization compare? If smooth filter initialization is consistently superior to pooling (even under certain controlled conditions), on a wider variety of datasets, tasks and architectures, it would indeed be a very interesting result.

---

### Official Review · AnonReviewer4 · 2018-11-12
**Not enough evidence to conclude much about pooling**

**Rating:** 4
**Confidence:** 4

**Review:**

It is often argued that one of the roles of pooling is to increase the stability of neural networks to
deformations. This paper presents empirical evidence to contest this assertion, or at least qualify it.

I appreciate empirical studies that question some of the widely accepted dogmas of deep learning.
From this point of view, the present paper is certainly interesting.

Unfortunately, the actual evidence presented is quite weak, and insufficient to draw far reaching
conclusions. An obvious objection is the authors only consider two datasets, and a very small number of
more or less standard pooling methodologies. The effect of pooling is evaluated in terms of cosine
similarlity, which is not necessarily a good proxy for the actual performance of a network.

A more serious issue is that they seem to very readily jump to unwarranted conclusions. For example,
the fact that stability to deformations (by which I necessarily mean the specific type of deformations
that they consider) tends to decrease in the middle layers of neural networks during training does not
mean that starting with a neural network with less stability would be better. Maybe some kind of
spontaneous coarse-to-fine optimization is going on in the network. Similarly, it is obvious that smoother
filters are going to lead to more stable representations. However, they might be less good at discriminative
tasks. Just because smoother filters are more stable does not automatically mean that they are more desirable.

Stability to deformations is an important but subtle topic in computer vision. For starters, it is difficult
to define what kind of deformations one wants to be insensitive to in the first place. A useful model would
likely incorporate some notion of deformations at multiple different length scales.

Just showing that one network is better than another wrt some arbitrarily defined simple class of deformations
with no reference to actual recognition performance, speed of training, or interpretation of the nature of
the deformations and the learned filters is not very convincing. I would particularly like to emphasize the
last point. I would really like to understand what pooling actually does, not just at the level of "if you
turn it off, then cosine similarity will decrease by this much or that much."

---

> ### Author Response · Authors · 2018-11-26
> **Thank you for your feedback**
>
> Thank you for your encouraging words regarding empirical studies that question some of the widely accepted dogmas of deep learning!
>
> We wish to clarify a few points and ask of you to clarify some of your comments if possible:
>
> “Just because smoother filters are more stable does not automatically mean that they are more desirable.”
> It seems that you concluded that we were claiming that more stability is always a desirable property. We have not asserted this and in fact have highlighted that often stability is *reduced* over the course of training. Further, as our title suggests, it is important not only to have “more deformation stability” but rather the “appropriate deformation stability”.
> In future we will try to make it clearer as to what we are asserting, and if you have any suggestions on how to improve this aspect we would greatly appreciate it.
>
> “Just showing that one network is better than another wrt some arbitrarily defined simple class of deformations with no reference to actual recognition performance, speed of training, or interpretation of the nature of the deformations and the learned filters is not very convincing.”
> Could you please clarify what you meant by this sentence. It is not clear what you think we are trying to convince you of when you say “this is not very convincing”.
> You assert that class of deformations is arbitrary. We spend the first few paragraphs of section 2.1 justifying the study of these deformations. It would be helpful for us if you could explain why this class of deformations still arbitrary to you.
>
> “I would really like to understand what pooling actually does.” It would be really helpful to us if you could expand on this and clarify what you are asking here.

---

### Official Review · AnonReviewer6 · 2018-11-15
**Serious empirical study, but somewhat unsurprising and expected conclusions.**

**Rating:** 5
**Confidence:** 5

**Review:**

This paper asks what is the role of pooling in the success story of CNNs applied to computer vision.
Through several experimental setups, the authors conclude that, indeed, pooling is neither necessary nor sufficient to achieve deformation stability, and that its effect is essentially recovered during training.

The paper is well-written, it is clear, and appears to be readily reproducible. It addresses an interesting and important question at the interface between signal processing and CNNs.

That said, the paper does not produce any clear novel results. It does not provide any theoretical result, nor any new algorithm. Its contributions consist of three empirical studies, demonstrating that (i) the benefits of pooling in terms of deformation stability can be achieved through supervised learning the filters instead (sec 3), (ii) the mechanism to obtain stability through learning essentially consists on reducing the bandwidth of (some) filters (sec4), and (iii) that this mechanism is data-dependent (sec 5). None of these studies strike the reviewer as particularly revealing. Moreover, the reviewer felt that the authors could have built on those findings to ask (and hopefully answer) a few interesting questions, such as:
-- Nowhere in the paper there is a discussion about critical Nyquist sampling and the need to reduce the bandwidth of a signal prior to downsampling it in order to avoid aliasing. Average pooling provably does it, and learnt filters do it provided they indeed become bandlimited. What are the links between deformation stability and the ability to avoid aliasing?
-- How many lowpass antialiasing filters are needed per layer to provide sufficient stability?
-- Also, the authors should relate this study with similar works that do the same in speech (e.g. https://www.isca-speech.org/archive/Interspeech_2018/abstracts/1371.html).

In conclusion, my impression is that this paper requires a major iteration before it can be of widespread interest to the community. I encourage the authors to think about the above points.

---

> ### Author Response · Authors · 2018-11-26
> **Thank you for your feedback**
>
> Thank you for your kind feedback about the writing and the importance of the question being addressed.
>
> While we agree the results are not particularly surprising in retrospect, reading the literature on pooling, we have not seen learned smooth filters as a proposed mechanism for deformation stability and thought that these results may be of interest to the community trying to understand convolutional networks and deformation stability.
>
> Further, we believe this work gives us an important bit of information on the topic of building inductive biases into architecture. Our work shows that a common architectural decision (pooling), long believed to be helpful in conferring a particular inductive bias (stability to deformation) was not actually necessary and that the inductive bias built in was being “overridden” by the learning process.
>
> Thank you for suggesting we discuss aliasing. This indeed looks like an important direction in which to expand this work. Also, thank you for pointing out the reference “Impact of Aliasing on Deep CNN-Based End-to-End Acoustic Models”, we were unaware of this work and it seems very relevant.

---

### Meta-Review · Area_Chair1 · 2018-12-17
**Extra iteration needed**

**Confidence:** 5
**Recommendation:** Reject

**Metareview:**

This paper studies the role of pooling in the success underpinning CNNs. Through several experiments, the authors conclude that pooling is neither necessary nor sufficient to achieve deformation stability, and that its inductive bias can be mostly recovered after training.

All reviewers agreed that this is a paper asking an important question, and that it is well-written and reproducible. On the other hand, they also agreed that, in its current form, this paper lacks a 'punchline' that can drive further research. In words of R6, "the paper does not discuss the links between pooling and aliasing", or in words of R4, "it seems to very readily jump to unwarranted conclusions". In summary, the AC recommends rejection at this time, and encourages the authors to pursue the line of attack by exploring the suggestions of the reviewers and resubmit.